# Canadian Healthcare Professionals’ Views and Attitudes toward Risk-Stratified Breast Cancer Screening

**DOI:** 10.3390/jpm13071027

**Published:** 2023-06-21

**Authors:** Julie Lapointe, Jean-Martin Côté, Cynthia Mbuya-Bienge, Michel Dorval, Nora Pashayan, Jocelyne Chiquette, Laurence Eloy, Annie Turgeon, Laurence Lambert-Côté, Jennifer D. Brooks, Meghan J. Walker, Kristina Maria Blackmore, Yann Joly, Bartha Maria Knoppers, Anna Maria Chiarelli, Jacques Simard, Hermann Nabi

**Affiliations:** 1Oncology Division, CHU de Québec-Université Laval Research Center, 1050, Chemin Sainte-Foy, Québec City, QC G1S 4L8, Canada; julie.lapointe@crchudequebec.ulaval.ca (J.L.); jean-martin.cote@crchudequebec.ulaval.ca (J.-M.C.); cynthia.mbuya-bienge.1@ulaval.ca (C.M.-B.); michel.dorval@crchudequebec.ulaval.ca (M.D.); jocelyne.chiquette.med@ssss.gouv.qc.ca (J.C.); annie.turgeon@crchudequebec.ulaval.ca (A.T.); laurence.lambert-cote@crchudequebec.ulaval.ca (L.L.-C.); jacques.simard@crchudequebec.ulaval.ca (J.S.); 2Department of Social and Preventive Medicine, Faculty of Medicine, Université Laval, 1050, Av de la Médecine, Québec City, QC G1V 0A6, Canada; 3Faculty of Pharmacy, Université Laval, 1050, Av de la Médecine, Québec City, QC G1V 0A6, Canada; 4CISSS de Chaudière-Appalaches Research Center, 143 Rue Wolfe, Lévis, QC G6V 3Z1, Canada; 5Department of Applied Health Research, Institute of Epidemiology and Healthcare, University College London, Gower Street, London WC1E 6BT, UK; n.pashayan@ucl.ac.uk; 6CHU de Québec—Université Laval, 1050, Chemin Sainte-Foy, Québec City, QC G1S 4L8, Canada; 7Programme Québécois de Cancérologie, Ministère de la Santé et des Services Sociaux, 1075, Chemin Sainte-Foy, Québec City, QC G1S 2M1, Canada; laurence.eloy@msss.gouv.qc.ca; 8Dalla Lana School of Public Health, University of Toronto, 155, College Street, Toronto, ON M5T 3M7, Canada; jennifer.brooks@utoronto.ca (J.D.B.); meghan.walker@utoronto.ca (M.J.W.); anna.chiarelli@cancercare.on.ca (A.M.C.); 9Cancer Care Ontario, Ontario Health, 525, University Avenue, Toronto, ON M5G 2L3, Canada; kristina.blackmore@cancercare.on.ca; 10Centre of Genomics and Policy, McGill University, 740, Ave Penfield, Montreal, QC H3A 0G1, Canada; yann.joly@mcgill.ca (Y.J.); bartha.knoppers@mcgill.ca (B.M.K.); 11Human Genetics Department and Bioethics Unit, Faculty of Medicine, McGill University, 3647, Peel Street, Montreal, QC G1V 0A6, Canada; 12Department of Molecular Medicine, Faculty of Medicine, Université Laval, 1050, Avenue de la Médecine, Québec City, QC G1V 0A6, Canada

**Keywords:** breast cancer, health policies, health professionals, implementation, national health systems, perspectives, personalized screening, risk-based strategy, screening programs

## Abstract

Given the controversy over the effectiveness of age-based breast cancer (BC) screening, offering risk-stratified screening to women may be a way to improve patient outcomes with detection of earlier-stage disease. While this approach seems promising, its integration requires the buy-in of many stakeholders. In this cross-sectional study, we surveyed Canadian healthcare professionals about their views and attitudes toward a risk-stratified BC screening approach. An anonymous online questionnaire was disseminated through Canadian healthcare professional associations between November 2020 and May 2021. Information collected included attitudes toward BC screening recommendations based on individual risk, comfort and perceived readiness related to the possible implementation of this approach. Close to 90% of the 593 respondents agreed with increased frequency and earlier initiation of BC screening for women at high risk. However, only 9% agreed with the idea of not offering BC screening to women at very low risk. Respondents indicated that primary care physicians and nurse practitioners should play a leading role in the risk-stratified BC screening approach. This survey identifies health services and policy enhancements that would be needed to support future implementation of a risk-stratified BC screening approach in healthcare systems in Canada and other countries.

## 1. Introduction 

Breast cancer (BC) is the most common cancer worldwide, with 2.3 million people diagnosed and 685,000 deaths in 2020 [1]. In Canada, it has been estimated that one in eight women will develop BC in their lifetime [2]. To enable early detection of this disease, the Canadian Task Force on Preventive Health Care recommends a mammography screening every 2 or 3 years for women aged 50 to 74 years [3]. Mammography screening has been associated with an approximately 20% decreased risk of BC mortality [4]. However, mammography screening can also have adverse consequences, such as overdiagnosis, false-positives and false-negatives, which lead to unnecessary procedures and can create psychological stress and anxiety, as well as an unnecessary burden on healthcare resources [4]. Currently, important research projects in Canada [5], the United States (US) [6] and Europe [7] are exploring the possibility of implementing a risk-stratified approach to BC screening [8]. In fact, evidence of the cost-effectiveness of a risk-stratified approach is mounting [9,10,11,12]. In contrast to current guidelines, which mainly use age as the eligibility criterion for BC screening, a risk-stratified approach would use family history of cancer; breast density; age, hormonal and lifestyle factors; and genetic information (including rare highly penetrant variants and common variants combined as a polygenic risk score [PRS]) to determine women’s risk categories [13]. The optimal screening strategies or interventions would then be based on these risk categories [14].

While a risk-stratified approach to BC screening seems promising [9,14,15,16], its implementation requires the buy-in of all stakeholders, particularly from those who would have a prominent role in the approach, such as healthcare professionals (HCPs). Indeed, HCPs would have to communicate with patients about their specific breast cancer risk, the potential benefits and harms of a risk-stratified approach to screening, the notion of a PRS and current screening recommendations based on their patients’ risk category [17]. Understanding HCPs’ views and attitudes regarding this approach will solidify the foundation for efficient implementation strategies [18,19].

Prior qualitative studies [20,21,22,23,24,25,26,27,28] explored HCPs’ views and attitudes toward the implementation of risk-stratified approaches to BC screening. However, to our knowledge, only two quantitative surveys examined this specific subject, one amongst Spanish HCPs (n = 220) [29] and one amongst U.K. general practitioners (n = 109) [30]. These studies reported that HCPs were concerned about the anxiety this approach could bring for women found to be at high risk [20,21,22,23,25,26]. Some HCPs were also concerned about the time and human resources that would be required to implement it [20,22]. Finally, HCPs voiced reservations related to the prospect of decreasing BC screening frequency for women found to be at low risk [20,24,27,30]. Despite those concerns, overall, HCPs viewed a risk-stratified BC screening approach as an important step to increase the efficiency and effectiveness of current BC screening programs [20,22,27,28,29,30]. 

This study explores Canadian HCPs’ views and attitudes regarding (1) BC screening recommendations based on individual risk categories, (2) their scope of practice and perceived readiness and comfort related to the possible implementation of this approach, (3) aspects of the healthcare system that should be enhanced to facilitate a risk-stratified approach to screening, and (4) the professional group that should be leading the integration of the risk-stratified BC screening approach. For this exploratory work, we had no prespecified hypotheses. These results will help identify potential barriers in the integration of a risk-stratified BC screening approach in Canada and other countries.

## 2. Materials & Methods

### 2.1. Study Design

Details on our methodology were previously published [31]. In brief, this study is part of a major Canadian research project entitled PERSPECTIVE I&I (Personalized Risk Assessment for Prevention and Early Detection of BC: Integration and Implementation), which aims to generate evidence to support the potential implementation of a risk-stratified BC screening approach in the Canadian healthcare system [5]. For this cross-sectional study, we disseminated (between November 2020 and May 2021) an online questionnaire through the networks of the Canadian PERSPECTIVE I&I study co-investigators and through the newsletters of 18 healthcare professional associations (see Appendix A). Eligible respondents comprised all persons who self-identify as HCPs and who were interested in providing their opinion and expectations regarding the implementation of a risk-stratified BC screening approach. The questionnaire was anonymous and self-administered. In the introductory section of the questionnaire, respondents were informed that their consent was implied by their completion of the questionnaire. The CHU de Québec–Université Laval’s Research Ethics Board approved this study (registration number: F9-55772).

### 2.2. Data Collection Tool

Based on the previous literature [13,21,28,32,33], the 17-item questionnaire (see Appendix A) was developed in French and English by a team of clinicians, epidemiologists and social scientists. The questionnaire was pilot-tested with seven HCPs, being careful to choose HCPs from the targeted population that were not involved in our study. The questionnaire first presented the elements of a risk-stratified BC screening approach using text and pictograms. Questions collected information on HCPs’ attitudes to the following issues: attitudes toward BC screening recommendations in a context of risk stratification (1 question with 6 statements),attitudes toward their role and scope of practice within a risk-stratified BC screening approach (1 question with 5 statements),views toward the necessary enhancements to the healthcare system required if such an approach were implemented (1 question),views toward the professional group that should play a role if risk-stratified BC screening were implemented (1 question).

In developing our set of questions, we adhered to a definition of *attitudes* that comprised three characteristics: “(a) a mental state—conscious or unconscious; (b) a value, belief, or feeling; and (c) a predisposition to behavior or action [34]. Similarly, we adhered to a definition of *views* as a point of view or a “position or perspective from which something is considered or evaluated” [35]. Finally, we asked six sociodemographic questions (i.e., gender, profession, main medical specialty, practice seniority, institution of practice and practice region). The online platform of our questionnaire was provided via Research Electronic Data Capture (REDCap) [36]. 

### 2.3. Statistical Analyses

For data analyses, medical specialties were categorized into three groups: “Family medicine/Primary care”, “Oncology” and “Other”. Practice seniority was categorized as follows: less than 5 years, between 5 and 14 years, between 15 and 25 years and more than 25 years. Practice regions were categorized as “Province of Québec”, “Province of Ontario” and “Other Canadian provinces and territories”. Analyses of questions related to the enhancements of the healthcare system and the designation of leading roles were stratified by region to reflect differences in provincial healthcare systems. The questionnaire concluded with one open-ended question asking respondents whether they had any comments or suggestions. 

Descriptive statistics were used to summarize responses. Chi-square tests were used to explore whether respondents’ attitudes differed according to sociodemographic and professional status. Dummy variables were created for missing responses. Analyses using listwise deletion of missing variables were also conducted as sensitivity analyses and were finally chosen for conducting exploratory analyses [37]. All tests were two-sided with a 0.05 level of significance. All statistical analyses were performed using SAS software, Version 9.4 (Copyright© 2016 by SAS Institute Inc., Cary, NC, USA).

Responses to the open-ended question were reviewed and a summary of responses was provided with illustrative quotes. 

## 3. Results

The questionnaire was completed by 593 respondents, and 453 (76.4%) of them completed all questions. As presented in Table 1, 432 (93.5%) respondents were female, 103 (22.3%) were physicians, and 323 (69.7%) were nurses (i.e., nurses or nurse practitioners). The distribution of medical specialities was as follows: family medicine/primary care (36.1%), oncology (12.8%) and other (51.1%). Other medical specialties included internal medicine, surgery, emergency, palliative care, public health medicine, radiology and obstetrics—gynecology. The three most frequent practice settings were academic hospitals (28.9%), community hospitals (21.3%) and community health centers (17.0%). Finally, respondents were mostly from the province of Québec, which had 82.9% of respondents, followed by Ontario at 10.1% and other Canadian provinces and territories at 7.0%. 

Close to 90% of respondents agreed with the recommendations of increasing the frequency of screenings and initiating BC screening at a younger age for women found to be in the high-risk group (Figure 1). However, only 9% agreed with the recommendation of not offering BC screening for women in a very low-risk group. Table 2 presents respondents’ choice regarding the professional group that should play a role if BC screening based on personalized risk stratification were implemented. Overall, primary care physicians and nurse practitioners were the groups of professionals most frequently recommended to play a role. Significant differences in terms of frequency of selected options are seen when data are stratified between Québec and other Canadian provinces. Access to a primary care physician is the most frequently endorsed aspect requiring enhancement to support the implementation of a risk-stratified approach (Table 3). Respondents’ answers were significantly different when stratified according to their region of practice. Indeed, the top three enhancement priorities for respondents from Québec were access to a primary care physician, number of nurse practitioners and access to breast screening (e.g., mammogram or MRI). The top three enhancement priorities for respondents from the other regions of Canada were access to breast screening, access to a nurse or nurse practitioner and number of genetic counselors. We conducted an exploratory analysis among the subgroups of physicians and nurses to appraise if the top three enhancement priorities would differ across regions within the same profession. As shown in Table 4, the top three priorities also differ within the same profession across regions. 

The majority (i.e., >60%) of respondents believed that “discussing the advantages and limitations of personalized BC risk assessment”, “collecting patient information required to perform a BC risk assessment” and “explaining to patients the difference between a risk of developing BC and a diagnosis of BC” might be part of their role (Figure 2). Among those who answered that it was their role, between 54.8 and 73.7% reported being comfortable with these roles. The role of “discussing the results of a BC risk assessment with a patient” was the least frequently endorsed one, with 51.3% of respondents endorsing it. However, 57.2% reported being comfortable with this role. Exploratory analyses revealed that respondents with a medical specialty in family medicine or primary care and those operating a practice in a family health team were significantly more likely to endorse their role in BC risk assessment and communication (see Appendix A). No clear response pattern was observed for the level of comfort associated with these roles (see Appendix A). 

A total of 61 respondents wrote comments or suggestions to the open-ended question. Expression of support for risk-stratified BC screening was present in 26 free-text responses (43%). Other comments concerned the prominent role nurses could play, the importance of access to training, the engagement of patients in an active and central role and the practical considerations of implementing this approach in the current healthcare system (see Appendix A). The following are some illustrative quotes from respondents: 

“Am very glad that there is enlightened exploration of routine breast screening practices with risk stratification. I personally have always challenged the recommendations and practices. We also need improved technology to assess breast health as all breast sizes and density are not equal. There are insufficient vertical MRIs available for routine screenings. I eagerly look forward to improved risk assessment and decision making supports”.“Moving from routine to individualized screening is a good idea, but risk assessment cannot be left to family physicians again. Some people do not have family physicians and family physicians already provide primary, secondary and even tertiary care. All specialties offload their follow-ups to family physicians. Adding a risk calculation to family physicians assumes that some will not have this calculation because of lack of time, because of other health problems to discuss or simply because they do not have a family physician. Women who may be at high risk could end up not being screened…”“Shared decision making should be central to this modified program. The risk-based approach is interesting but will need to be the subject of randomized trials to properly assess the risks and benefits. In all cases, the shared decision must be put at the heart of the discussions, which is far from being the case at present”.

## 4. Discussion

This study surveyed Canadian HCPs about their views and attitudes regarding the integration of a risk-stratified BC screening approach into Canada’s current provincial healthcare systems. A vast majority of HCPs supported the recommendations of increasing the frequency and initiating BC screening at a younger age for women found to be in a high-risk group. That support dropped substantially for the recommendation of less frequent and delayed BC screening for women in a low-risk group. Furthermore, respondents did not support the recommendation of not offering BC screening to women in a very low risk group. Respondents identified structural enhancements needed to support future implementation of a risk-stratified BC screening approach in some Canadian jurisdictions, including access to a primary care physician, the number of nurse practitioners and access to breast screening (e.g., mammogram, MRI).

This response pattern, i.e., positive views about increasing screening for women at high risk and concerns about reducing the frequency or not offering BC screening for women at low risk, was reported in several qualitative studies conducted among HCP populations [24,27] and one quantitative study conducted among general practitioners in U.K. [30]. This response pattern was also reported in previous studies conducted among women from the general population [32,38,39,40]. Notably, not offering BC screening to women at very low risk has never been a suggested recommendation of PERSPECTIVE I&I, which is our implementation and integration study [5]. The possibility of offering less frequent BC screening to women at low risk is viewed as a potentially contentious public health issue [24]. To support the debate around this specific issue, it would be important to garner robust evidence about and clearly communicate the benefit–harm trade-offs of decreasing screening frequency for women found to be at low risk of BC (reducing radiation exposure, overdiagnoses, detection of false-positive cases and anxiety of waiting for the mammogram results) [9,12,16,41]. Moreover, these communication efforts should be multi-pronged, led by credible sources and supportive of a dialogue with all stakeholders, including service users [21,24]. Indeed, the importance of shared decision-making between patients and healthcare providers in choosing a BC screening strategy was expressed in some of the free-text responses of our respondents and is regarded as an important consideration in current Canadian screening guidelines [3]. 

To our knowledge, this is the first report on how the geographical region of practice in Canada impacts the views of HCPs regarding the structural and systemic enhancements required to implement a risk-stratified BC screening approach. Canada has a population of approximately 39 million [42] and is divided into ten provinces and three territories, with Québec and Ontario being the most populous provinces, with estimated populations of 8 and 14 million, respectively. Since 1966, Canada has implemented a universal health care system, known as Medicare, that is managed by each province and territory [43]. This system means that each jurisdiction determines the medical acts covered by their health care plan [44]. We were not surprised by Québec respondents’ top choice of “access to a primary care physician”. For several years, this province has had the highest proportion of residents without a regular health care provider (i.e., 21.5% in 2019), while Ontario’s proportion of residents without a regular health care provider was 9.5% in 2019 [45]. Notably, we are acutely aware that these latter estimates may have shifted since the COVID-19 pandemic. We hypothesize that differences between the geographical regions of practice regarding the number of genetic counselors (7.4% for respondents from Québec, compared to 11.6% for respondents from other regions in Canada) could be at least partly explained by the fact that Ontario and British Columbia already have high-risk BC screening programs offering genetics counseling, testing and/or enhanced screening strategies [46,47]. The presence of these clinics may expose or familiarize respondents with the role of genetic counselors relative to regions where such clinics are not present. Our respondents’ views regarding the relative importance of medical training in the implementation of risk-based BC screening seems to differ from the views of HCPs practicing in Spain [29] and the U.K. [30]. Both the Spanish and U.K. respondents ranked training of professionals as the most important consideration for implementation, while our respondents ranked medical training in sixth position, after aspects related to access to primary care and nurses, number of nurses, access to breast screening and time allocation for appointments. This major difference in response pattern is another demonstration that it is necessary to collect the views of professionals practicing in regions where implementation is being considered.

There seems to be a consensus regarding the designation of primary care physicians and nurse practitioners as groups of HCPs who should play a leading role if risk-stratified BC screening were implemented. This designation is echoed in other studies [17,20,22,24,48]. However, this assumption raises concerns about the pressure on professionals’ workloads that this new role would add in the context of already extremely limited resources [20,21,22,29,49]. Solutions proposed by these later studies include use of result letters, availability of a helpline, adapted communication tools, proper planning of the BC screening and care pathways, the possibility to refer women found to be at high risk to specialists or special clinics, centralized public health programs, proper HCP training, general population education and awareness campaigns. As mentioned in the free-text responses of one of our respondents, it would be important to avoid offloading these new roles and responsibilities to primary care HCPs. 

Our result showing that “collecting patient information required to perform a BC risk assessment” was the most frequently endorsed role (67.8%) and the highest in terms of how comfortable professionals would be (73.7%) is in line with previous studies collecting HCPs’ attitudes toward the integration of genomics into clinical management [50]. Also, the comfort levels expressed for the roles of “discussing the advantages and limitations of personalized BC risk assessment” and “discussing the results of a BC risk assessment with patient” are similar to results reported by general practitioners [51,52], as well as being only slightly lower than results reported by medical oncologists [53] for similar roles in managing genomic information in practice. However, it is important to note that a sizeable proportion of our study respondents were ambivalent as to whether they would be comfortable with the proposed roles required within a risk-stratified BC screening approach. It would, thus, be important that healthcare decision makers lead a thorough consultation phase with professional associations and professional representatives in the process of implementing risk-stratified BC screening in order to identify actions, guidelines and resources to put in place for HCPs to feel comfortable in endorsing these new roles. Also, this consultation phase should be followed and/or informed by feasibility and pilot testing of different implementation approaches [24]. 

## 5. Strengths and Limitations

To our knowledge, this survey has the largest sample size of HCPs providing their views and attitudes toward a risk-stratified BC screening approach. The sample size was not only large, but also varied according to important parameters, such as profession, seniority and medical specialty. Our recruitment strategy was multi-pronged and supported by several professional associations and healthcare institutions. It is, however, important to recognize that the region of practice of our respondents was unevenly distributed, with a high proportion of respondents being from Québec, few being from the other Canadian provinces, and none being from the territories. The situation is mostly explained by the fact that we succeeded in gaining the support of nine professional associations based in Québec, four that were pan-Canadian and five based in Ontario or other Canadian provinces. Other studies will, thus, be needed to collect the representative views and attitudes of HCPs from all Canadian jurisdictions. Also, in developing our bilingual questionnaire, the question about which professional group should play a role in a risk-stratified BC screening approach presented a slightly different response scale between the French and English versions. The English version proposed two more choices than the French version. This issue limited our ability to compare the views of HCPs on this specific aspect. Fortunately, respondents were invited to check all choices they believed would apply; thus, we were able to analyse and compare the proportions for each of the other response choices. 

## 6. Conclusions

This study on HCPs’ views and attitudes contributes to the developing body of knowledge used to support future implementation of risk-stratified BC screening in Canadian healthcare systems, as well as in other countries’ systems. Our respondents underscored the health services and policy enhancements that would support their practice, as well as the efficiency of the approach. However, it is important that future research efforts collect data from representative HCP populations as our results showed that the priority of some enhancements were specific to the geographical region of practice. Finally, while there was general support for the recommendation of increasing BC screening for women at high risk, that was not the case for the recommendation of decreasing BC screening for women found to be at lower risk. If the goal is to balance the distribution of BC screening services in favor of those who would benefit the most, there is a need for resources to support communication and shared decision-making among patients and HCPs on the potential benefits and harms of BC screening according to the different risk categories. 

## Figures and Tables

**Figure 1 jpm-13-01027-f001:**
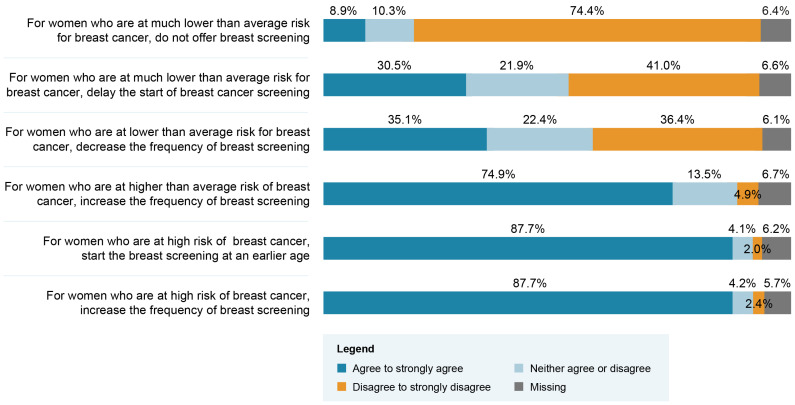
Healthcare professionals’ attitudes regarding possible breast cancer (BC) screening recommendations related to different risk categories (N = 593).

**Figure 2 jpm-13-01027-f002:**
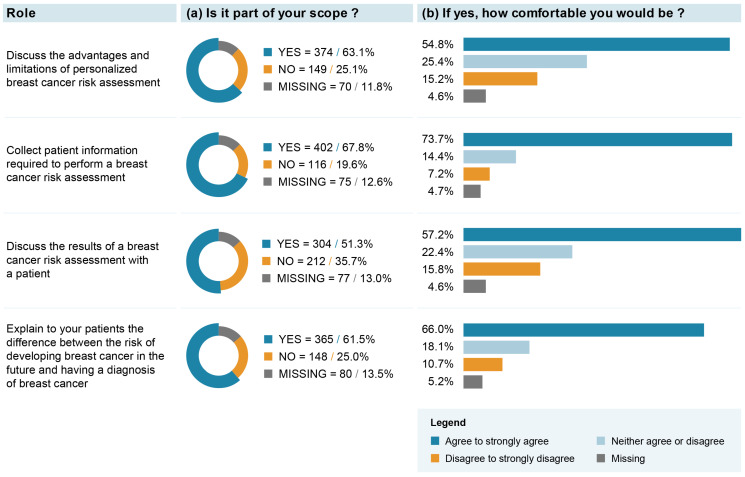
Healthcare professionals’ attitudes toward their potential role in integration of breast cancer (BC) screening based on individual risk level.

**Table 1 jpm-13-01027-t001:** Respondents’ characteristics (N = 593).

	n (%)
**Gender**	
Women	432 (93.5)
Men	30 (6.5)
[Missing data/Prefer not to answer]	[131]
**Profession**	
Physician	103 (22.3)
Nurse	323 (69.7)
Other (1)	37 (8.0)
[Missing data]	[130]
**Medical specialty**	
Family medicine/Primary care	167 (36.1)
Oncology	59 (12.8)
Other (2)	236 (51.1)
[Missing data]	[131]
**Number of years of practice**	
<5 years	58 (12.5)
5–14 years	135 (29.2)
15–25 years	113 (24.4)
>25 years	157 (33.9)
[Missing data]	[130]
**Region of practice**	
Province of Québec	377 (82.9)
Province of Ontario	46 (10.1)
Other Provinces (3)	32 (7.0)
[Missing data]	[138]
**Practice Setting**	
Academic Hospital	133 (28.9)
Community Hospital	98 (21.3)
Community health center	78 (17.0)
Family health team/group/network	75 (16.3)
Private clinic	25 (5.4)
Other (4)	51 (11.1)
[Missing data]	[133]

(1) Other professions include genetic counsellor, physiotherapist, occupational therapist, medical imaging, researcher and technologist; (2) other medical specialties include internal medicine, surgery, emergency, palliative care, public health medicine, radiology and obstetrics—gynecology; (3) Other provinces included British Columbia (n = 5), Alberta (n = 7), Manitoba (n = 7), Saskatchewan (n = 3), Prince Edward Island (n = 2), New Brunswick (n = 2), Nova Scotia (n = 1) and Newfoundland and Labrador (n = 5). None of territories were represented among our respondents. (4) Other practice settings include intensive care unit, nurse practitioner led clinic, nursing home, public health agency and research center.

**Table 2 jpm-13-01027-t002:** Healthcare professionals’ views on which professional groups should play a role if breast cancer (BC) screening based on personalized risk assessment were implemented (respondents could check all that applied).

	Entire Group n (%)	Province of Québec n (%)	Other Canadian Provinces n (%)	*p*-Value
Primary care physician	450 (24.7)	338 (26.3)	75 (20.4)	0.02
Nurse practitioner	425 (23.3)	318 (24.7)	67 (18.3)	0.01
Genetic counsellor	235 (12.9)	163 (12.7)	47 (12.8)	0.94
Nurse navigator	231 (12.7)	162 (12.6)	49 (13.4)	0.70
Geneticist	203 (11.1)	158 (12.3)	29 (7.9)	0.02
Radiologist	190 (10.4)	137 (10.6)	36 (9.8)	0.64
Other ^†^	87 (4.8)	11 (0.9)	64 (17.4)	<0.01

^†^ Other includes professionals such as surgeons and oncologists.

**Table 3 jpm-13-01027-t003:** Healthcare professionals’ views on most important aspects their provinces’ healthcare systems should enhance to implement breast cancer (BC) screening based on personalized risk assessment (respondents were invited to check top three).

	Entire Group n (%)	Province of Québec n (%)	Other Canadian Provinces n (%)	*p*-Value
Access to a primary care physician	229 (15.6)	187 (16.9)	25 (11.1)	0.03
Number of nurse practitioners	189 (12.9)	148 (13.4)	25 (11.1)	0.35
Access to breast screening (e.g., mammogram, MRI)	177 (12.0)	133 (12.0)	28 (12.4)	0.87
Time allocated to a patient-physician appointment	162 (11.0)	117 (10.6)	25 (11.1)	0.82
Access to a nurse or nurse practitioner	155 (10.6)	118 (10.7)	26 (11.6)	0.70
Medical training	152 (10.3)	125 (11.3)	15 (6.7)	0.04
Number of primary care physicians	118 (8.0)	82 (7.4)	25 (11.1)	0.06
Number of genetic counsellors	113 (7.7)	74 (6.7)	26 (11.6)	0.01
Time allocated to a patient-nurse practitioner appointment	71 (4.8)	50 (4.5)	13 (5.8)	0.42
Number of geneticists	46 (3.1)	36 (3.3)	6 (2.7)	0.64
Remuneration of healthcare professionals	34 (2.3)	26 (2.4)	7 (3.1)	0.51
None, I believe the healthcare system is ready	6 (0.4)	6 (0.5)	0 (0.0)	0.27
Other	17 (1.2)	2 (0.2)	4 (1.8)	<0.01

**Table 4 jpm-13-01027-t004:** Physicians and nurses’ views on most important aspects their provinces’ healthcare system should enhance to implement breast cancer (BC) screening based on personalised assessment (respondents could select up to three priorities).

	Physicians		Nurses	
	Québec (n = 76)	Other Provinces (n = 23)	Québec (n = 276)	Other Provinces (n = 46)
Access to a primary care physician	21	10	151	10
Number of nurse practitioners	13	2	130	21
Access to breast screening (e.g., mammogram, MRI)	15	8	108	18
Time allocated to a patient-physician appointment	35	11	74	11
Access to a nurse or nurse practitioner	8	3	105	21
Medical training	39	3	78	12
Number of primary care physicians	24	11	51	11
Number of genetic counsellors	28	10	37	10
Time allocated to a patient-nurse practitioner appointment	6	1	43	11
Number of geneticists	14	2	16	3
Remuneration of healthcare professionals	5	4	20	3
None, I believe the healthcare system is ready	2	0	4	0
Other	2	1	0	2

Note: Cells underlined in light blue represent top three aspects requiring enhancement for each respondent subgroup.

## Data Availability

The data that support the findings of this study are available from the corresponding author upon reasonable request.

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
