# Peer review of "Canadian Healthcare Professionals’ Views and Attitudes toward Risk-Stratified Breast Cancer Screening"

_jpm, 2023, doi:10.3390/jpm13071027_

Round 1
Reviewer 1 Report
This paper adopts a quantitative approach to report the views and attitudes of healthcare professionals on risk-stratified breast screening in Canada.
There have been a number of studies that have used a qualitative approach to examine attitudes of health care professional view and attitudes of risk-stratified breast cancer screening but this has only been quantified in a couple of studies, this study uses a survey to gather views of healthcare professionals in Canada.
Methods
How did you identify the target professional and healthcare institutions? How many did you approach in each region?
You could include some additional information about the questionnaire development – did you use a conceptual framework? How were questions selected/devised – was there a larger pool of questions which were reduced to those included in the questionnaire after review by e.g. experts in the field, or by using focus groups amongst target respondents? Did you pilot the questionnaire – if so, who piloted the questionnaire and what were the results of that pilot?
Please add a copy of the questionnaire as an appendix.
Results
I feel that some of the figures (e.g. 2 & 3) would be better presented as Tables including both counts and percentages within each category and including actual p-values for comparisons.
Table 2 would be better sorted in some way e.g. those with the top responses.
Table 3 looks a little messy and could perhaps be presented in a better format.
Not sure if the supplementary material was included in the submission – I do not seem to be able to see it.
Please make sure to follow STROBE guidelines.
Some minor grammatical errors
Author Response
Many thanks for your review. Please find attached our response.

Reviewer 2 Report
Dear author
Thank you for the submission of your article to our journal. I respect that you conducted questionnaires to many HCPs and analyzed the results. However, the main point of this study is to recommend active breast cancer screening for high-risk cases, which is what medical professionals around the world are thinking. Conversely, many medical professionals believe that omitting breast cancer screening for low-risk cases is inappropriate in terms of avoiding breast cancer mortality. This study may be meaningful in that it examines the thoughts of MCPs in Canada. However, it is highly questionable whether this study will have any impact on breast cancer screening around the world.
This paper is well written with the exceptions of a few careless mistakes.
Author Response
Many thanks for your review. Please see attached document for our response.

Reviewer 3 Report
The manuscript examines attitudes and values of healthcare providers on risk-stratified breast cancer screening. The manuscript presents results of a survey conducted among approximately 450 self-identified healthcare providers.
The manuscript should present some clarifying information, to enhance the validity of the study. Lines 100-105 explain that an online questionnaire was disseminated through professional associations and healthcare institutions. Can the authors expand this section? How were these organizations chosen and how representative were they?
The manuscript should also present some information of the target number of healthcare providers and the response rate. Also, what were the other Canadian provinces that were represented in responses?
Llnes 105-107 - the sentence needs editing.
Other language issues were not major, a spell check and grammar check is needed.
Author Response

(The authors gave the same response as above.)

Round 2
Reviewer 2 Report
Dear author
Thank you for the re-submission of your article to our journal. In the primary review, I described that medical professionals all over the world agree with the conclusions of this study. In other words, the contents of this article are commonplace for many professionals involved in breast cancer screening. I checked the revised paper, but I didn't see any major changes in the essential content. Scientific papers should contribute to daily clinical practice by allowing many medical professionals to learn new knowledge by reading the papers.